# Response of Individual-Tree Aboveground Biomass to Spatial Effects in *Pinus kesiya* var. *langbianensis* Forests by Stand Origin and Tree Size

Chunxiao Liu [1,2], Yong Wu [1,2], Xiaoli Zhang [1,2], Hongbin Luo [1,2], Zhibo Yu [1,2], Zihao Liu [1,2], Wenfang Li [1,2], Qinling Fan [1,2] and Guanglong Ou [1,2,*]

1 Key Laboratory of State Administration of Forestry and Grassland on Biodiversity Conservation in Southwest China, Southwest Forestry University, Kunming 650233, China; liuchunxiao@swfu.edu.cn (C.L.); yongwu@swfu.edu.cn (Y.W.); karojan@swfu.edu.cn (X.Z.); luohongbin@swfu.edu.cn (H.L.); yuzhibo@swfu.edu.cn (Z.Y.); liuzihao@swfu.edu.cn (Z.L.); lwf0920@swfu.edu.cn (W.L.); fql18095045157@swfu.edu.cn (Q.F.)
2 Key Laboratory for Forest Resources Conservation and Utilization in the Southwest Mountains of China, Ministry of Education, Southwest Forestry University, Kunming 650233, China
* Correspondence: olg2007621@swfu.edu.cn

**Abstract:** To enhance forest carbon sequestration capacity, it is important to optimize forest structure by revealing the spatial effects of the aboveground biomass of individual trees, with particular emphasis on stand origin and tree size. Here, 0.3 ha clear-cut plots of *Pinus kesiya* var. *langbianensis* forest were selected in a typical plantation and natural stand. Then, the ordinary least squares model and spatial regression models were used to analyze the different responses between spatial position and individual tree biomass based on the stand origin and diameter at breast height (DBH) of the tree. Our study shows the following: (1) The spatial effect produced a stronger response in the natural stand than in the plantation. The amount of change in the adjusted R-squared ($\Delta_{Radj}^2$) of tree component totaled 0.34 and 0.57 for *Pinus kesiya* var. *langbianensis* and other trees in the natural stand, compared to only 0.2 and 0.42 in the plantation; (2) Spatial effects had a stronger impact on the accuracy of the fit for the crown ($\Delta R_{adj}^2 = 0.52$) compared to the wood and bark ($\Delta R_{adj}^2 = 0.03$) in the plantation, and there were no significant differences in the natural stand ($\Delta R_{adj}^2 = 0.42$, $\Delta R_{adj}^2 = 0.43$); (3) When DBH reached a certain size, the impact of spatial effect for the crown showed a significant change from positive to negative. The sizes of DBH were 19.5 cm, 14 cm and 34.6 cm, 19 cm for branches of *Pinus kesiya* var. *langbianensis* and other tree species in the plantation and natural stand, and were 20.3 cm and 31.4 cm for the foliage of *Pinus kesiya* var. *langbianensis*. Differences in stand structure led to varied responses in the biomass of tree components to spatial effects.

**Keywords:** *Pinus kesiya* var. *langbianensis*; spatial regression model; tree biomass; plantation; natural stand; competition threshold





## 1. Introduction

Forest ecosystems play a key role in mitigating climate change, conserving biodiversity, and maintaining the global ecological balance [1]. For example, biomass in forest ecosystems sequesters about $2.4 \pm 0.4$ Gt C per year globally [2], providing the important climate regulation service of eliminating and sequestering atmospheric carbon [3]. Enhancing forest biomass is particularly crucial for the sustainable development of human societies, as Collins et al. [4] predicted that the global climate will gradually warm in the coming decades. The enhancement of forest biomass requires scientific, rational, and sustainable forest quality improvement approaches.

Forest stand restructuring is an important method for forest quality improvement [5]. In particular, the optimization of forest stand spatial structure can ensure a sustainable



supply of wood products, stabilization of water and nutrient cycles, and reduction of the risk of fire, windthrow, and land degradation, as well as safeguarding the recreational and educational functions of forest areas [6]. Such adjustments involve optimizing the spatial position between trees, improving stand diversity and health, and so on [7]. The spatial relationship between trees constitutes the spatial structure, which is essential for understanding tree growth potential, competitive relationships, and ecological niche distribution [8]. The spatial dispersion exhibited by spatial relationships determines the spatial effects directly, influencing the distribution of ecological conditions such as light, soil moisture, and air temperature, thereby promoting or constraining tree growth [9]. In addition, spatial effects influence the pattern of tree biomass allocation—which largely depends on tree access to growth resources—mainly through competition, which manifests in the differentiation of stand structure [10]. Exploring and clarifying the spatial effects among trees is crucial for forest management oriented towards forest carbon sink enhancement.

In the past, forest managers were mostly focused on maximizing timber products (economic returns) and forestry scholars were mostly focused on the growth in tree diameter and tree height, as well as the responses to spatial effects between stems [11,12]. However, trees include crown (branches, foliage) growth in addition to timber, which primarily affects the sustainability of non-timber products (ecological) [13,14]. Forest structure and the proportion of tree size dimensions differ depending on management practices and objectives. However, it is difficult to accurately obtain fundamental data for different tree components and tree sizes due to the lack of clear-cut sample plots, especially in natural stands. This has resulted in few studies on the spatial effects of tree component biomass by tree size within stands of different origins.

Spatial effects influence photosynthesis through competition and collaboration among trees, as well as the distribution of light in forests. Though all components of a tree are capable of photosynthesis [15], the tree components differ in their rate of photosynthesis—and, thus, contribute to regulating climate change—due to differences in biomass as well as in chlorophyll content and sources of captured $CO_2$ [16,17]. A reasonable increase in crown biomass is as important as an increase in trunk biomass for ameliorating global climate change and global carbon storage. Therefore, studies on the spatial effects of individual trees cannot be limited to the trunk, but should also consider the crown.

Forest origin (natural and plantation) may be a factor influencing differences in the response of individual-tree biomass to spatial effects, due to differences in tree species complexity and tree size. The variations arising from distinct tree species compositions primarily stem from variations in the levels of both intraspecific and interspecific competition among trees. Some studies found differences in the complexity of stand structure and biodiversity in forests of different origins, even under near-natural stand management [7]. Dănescu, Albrecht, and Bauhus [18] showed that the complexity of stand attributes (structure and biodiversity) directly increases stand productivity or aboveground biomass. In addition, the diversity–productivity relationship (DPR) in forest ecosystems may be influenced by stand origin, since a simple understanding of DPR in controlled homogeneous environments may not reflect the situation in heterogeneous natural environments [19]. The actual effectiveness of ecological niche complementarity is also related to stand complexity due to complex stands tending to make better use of horizontal and vertical space [20].

Another issue that needs to be considered is that the spatial effects of trees may vary with diameter at breast height (DBH). Forest researchers have found that spatial patterns of tree location are strongly influenced by competition between neighboring trees [21]. Different DBHs' relative efficiency in utilizing the available resources can be evaluated using a size–growth relationship (SGR) [22]. The SGR moves from size symmetry to size asymmetry, and then to anti-size asymmetry, in the forest as the stand progresses [22,23]. However, SGR does not consider the effect of tree spatial patterns on tree size inequality [24], although Weiner, Stoll, Muller-Landau, and Jasentuliyana [24] pointed out that the degree of size asymmetry in competition is due to differences in spatial structure. This demonstrates that the spatial competition relationship of DBH differs under the influence of spatial effects.

Overall, optimization of forest spatial structure is the key to increasing forest carbon sinks and mitigating global climate change. To further explore the differences in the response of aboveground biomass of individual trees to spatial effects, this study selected a total of two clear-cut sample plots as the research object in Pu'er, Yunnan province, including one plantation and one natural stand. The ordinary least squares models (OLS), spatial lag model (SLM), spatial error model (SEM), spatial Durbin model (SDM), and geographically weighted regression model (GWR) were chosen to fit the tree component biomass (wood, bark, branches, foliage) and individual tree aboveground biomass of *Pinus kesiya* var. *langbianensis* and the other species of trees present. This study aimed to elucidate the differences in spatial effects between (1) the tree component biomass of individual trees; (2) stands of different origin; and (3) the total aboveground biomass of individual trees along with changes in tree size. This would reveal the patterns underpinning spatial effects in forests and provide a reference for the spatial structure adjustment of forest stands for the enhancement of stand biomass.

## 2. Data and Methods

### 2.1. Data

#### 2.1.1. Study Area

The study area was located in the city of Pu'er, Yunnan, southwestern China. The landscape of the district is mostly mountainous and hilly, with a subtropical monsoon climate zone and altitudes ranging from 337 to 3370 m above sea level. The plantation sample plot was situated in Simao district, between 22°29′ and 22°49′ N and 100°38′ and 100°56′ E. The *Pinus kesiya* var. *langbianensis* has the same spacing within the sample plots, and the plots are thinned regularly to promote good stand spatial structure. The natural stand sample plot is located in Mojiang county, between 23°03′ and 23°15′ N and 101°23′ and 101°38′ E (Figure 1), and is an undisturbed human interference forest stand.

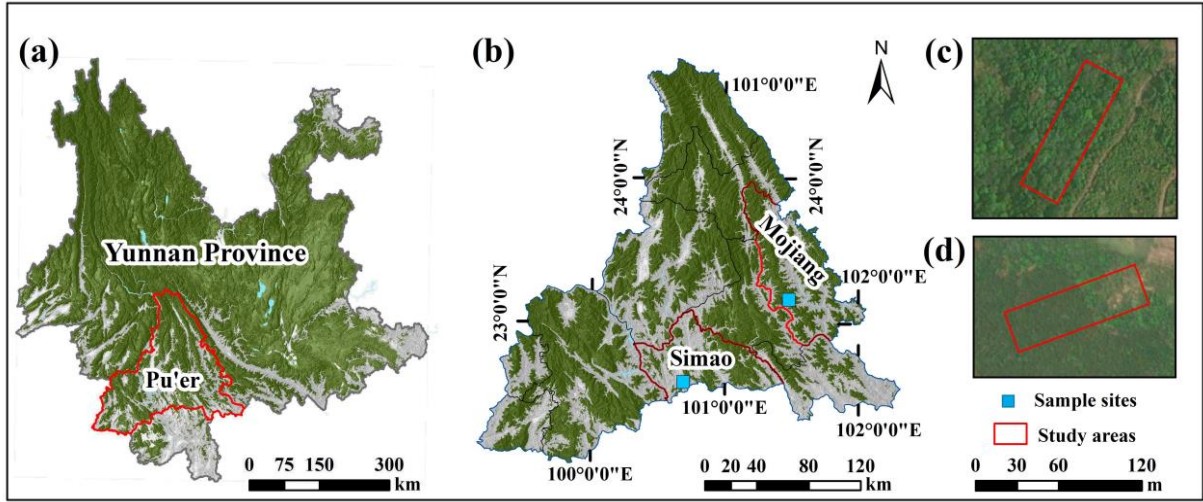

**Figure 1.** The location of clear-cutting sample plots. (**a**,**b**) show the location of the plantation sample plots in Simao district and the natural stand sample plots in Mojiang County of Pu'er City, Yunnan Province, China. (**c**) is a satellite image of the natural stand site, and (**d**) is a satellite image of the plantation site.

#### 2.1.2. Field Survey Data

The plantation sample site (100°41′46.63″ E–22°35′48.91″ N) has an average annual temperature of 19.3 °C and an average annual rainfall of 1442.2 mm. The natural stand sample site (101°29′14.40″ E–23°09′58.20″ N) has an average annual temperature of 18.2 °C and an average annual rainfall of 1322.7 mm. Both plantation and natural stand plots are located in the subtropical monsoon zone with warm, rainy summers and relatively dry winters. The soil types are both yellow loam and middle-aged forests. The site conditions

and stand structure are the same, and they can represent the same forest type in the region. To obtain basic data by tree component for individual trees, we set up a 30 × 100 m sample plot in both the plantation and natural origin stands. The coordinates of the living trees in the sample plot were recorded in detail, starting from the southwest corner of each sample plot (Figure 2), and then all the trees in the sample plot were cut down in April and May 2015. Comprehensive data on species, diameter at breast height (DBH), tree height (H), crown length (CL), and crown width (CW) were recorded for trees with a DBH of more than 5 cm (Figure 3). The plantation consisted of 358 trees with a total of 12 species. It was mainly dominated by *Pinus kesiya* var. *langbianensis* species, and there were also *Castanopsis delavayi*, *Schima wallichii*, *Ficus hispida*, and *Phyllanthus emblica* species growing in the forest stand. There were a total of 437 trees in the natural stand, with a total of 38 tree species. These were mainly *Pinus kesiya* var. *langbianensis*, *Schima wallichii*, *Eurya japonica*, *Quercus delavayi*, *Alnus nepalensis*, *Phyllanthus emblica*, and other species growing in the natural stand.

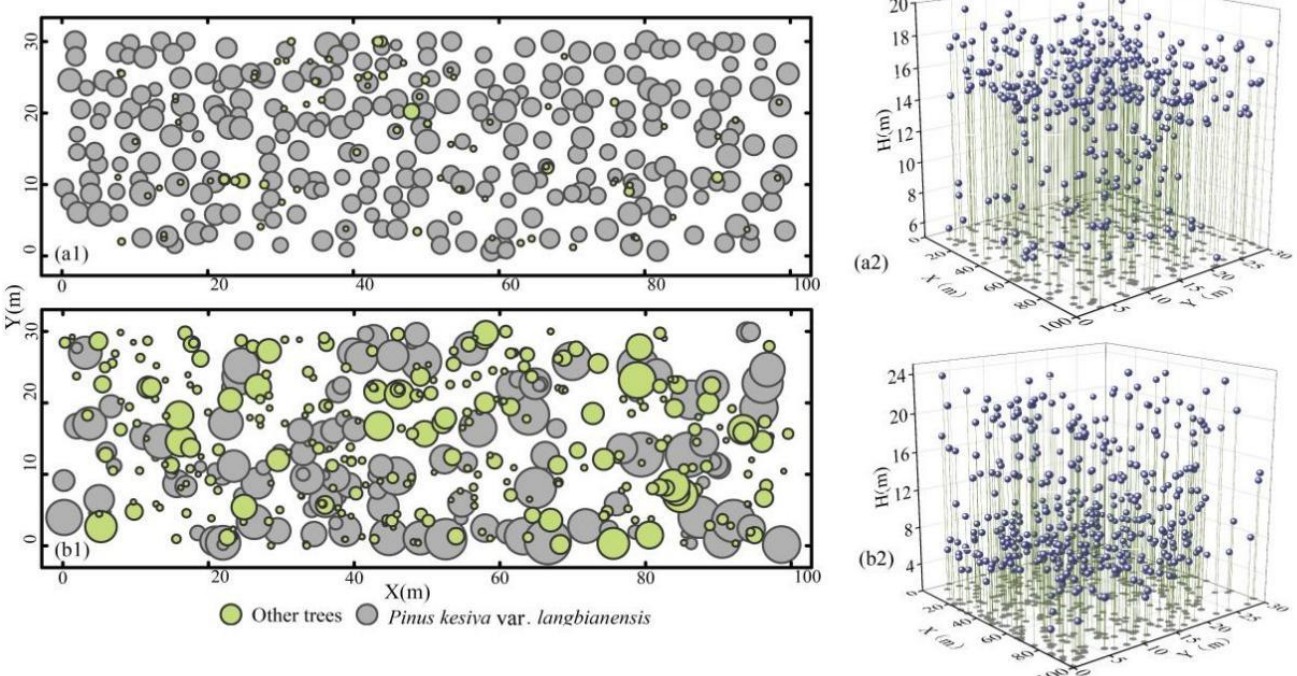

**Figure 2.** Sample plot schematic. (**a1**) is the DBH distribution in plantation, and (**b1**) is the DBH distribution in natural stand, where bubble size is proportional to DBH. (**a2**) is the distribution of H in the plantation, and (**b2**) is the distribution of H in the natural stand, where H (m) is the height of trees, while X (m) and Y (m) are the boundaries of sample plots.

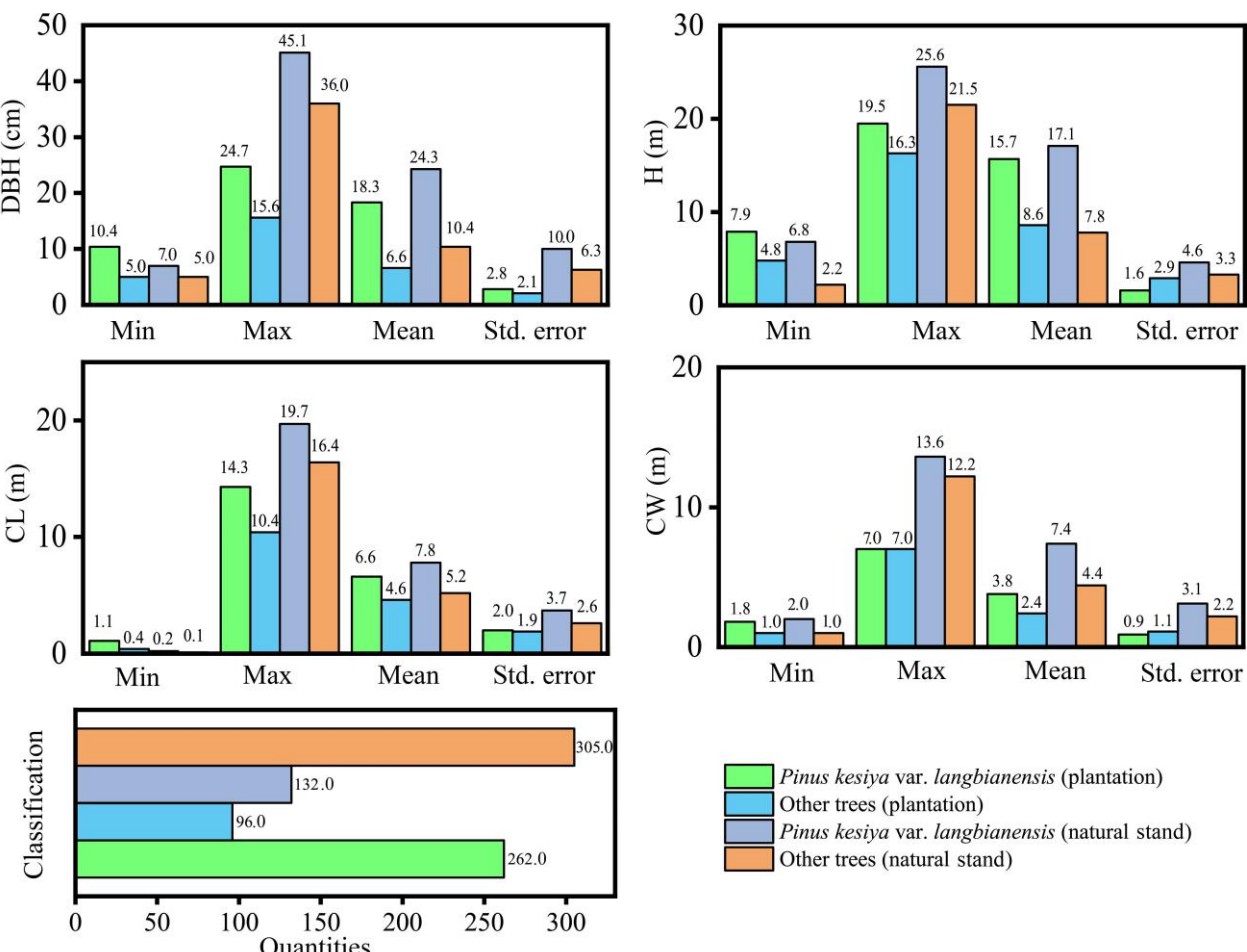

**Figure 3.** Basic information of sample plot.

*2.2. Methods*

2.2.1. Biomass Calculation

Biomass data for different tree components, including wood, bark, branches, foliage, and aboveground biomass, were obtained using weighting and bulk density methods [25]. The bulk density method uses the dry weight and volume of the sample to calculate the density and, thus, the biomass of the tree. The ratio of dry weight to volume is the density of the sample. The product of the density and volume of each section represents its biomass. The biomass of wood and bark was calculated using the bulk density method: cut the tree close to the ground, divide it into several segments according to a 2 m section, and measure the diameter with and without bark of each section. The volume of trunk and wood in each section was calculated according to the average section area method. Next, the felled wood was categorized into upper, middle, and lower layers to account for potential density variations in different parts of the stem. Specifically, samples were collected at 1/4, 1/2, and 3/4 points along the stem. The volume of the samples was subsequently determined using the drainage method, and the samples were ultimately dried in an oven at 105 °C until reaching a constant weight. Finally, the volume of each layer was distributed proportionally, dividing it into three equal segments based on the tree height. The biomass of wood and bark can be calculated based on the volume of each layer combined with the density of wood and bark calculated based on total dry weight and total volume:

$$w_i = \sum_{j=1}^{n} \rho_j \cdot V_j \tag{1}$$

where $w_i$ is the biomass of the wood or bark, $n$ is the layer numbers, and $\rho_j$ and $V_j$ are the density and volume of layer $j$, in which

$$V_k = \frac{A_{1k} + A_{2k}}{2} \cdot h_k \tag{2}$$

where $V_k$ is the volume of segment $k$, $A_{1k}$ and $A_{2k}$ are the upper and lower cross-sectional areas of segment $k$ (unit cm$^2$), and $h_k$ is the length of segment $k$ (unit m).

The biomass of branches and foliage was calculated using a weighted method. In the field, the branches (without leaves) and foliage of felled trees were fully weighted. Then, standard branches were selected according to the standard branch method, and the fresh weight of branches (without foliage) and foliage were measured. The standard branches were subjected to baking in an oven at 105 °C until a constant weight was achieved, following which the moisture content of branches and foliage was calculated. Finally, incorporating the water content and the proportion of the fresh weight of the standard branch can determine the biomass of the branches and foliage:

$$w_i = \frac{m_a - m_b}{m_a} \cdot m \tag{3}$$

where $w_i$ is the biomass of the branch or foliage, $m_a$ and $m_b$ are the fresh weight and dry weight of the sample, and $m$ is the total fresh weight of the branch or foliage.

Regarding aboveground biomass, it was calculated as the sum of the 4 aboveground tree components, namely, the wood, bark, branches, and foliage:

$$w_i = w_a + w_b + w_c + w_d \tag{4}$$

where $w_i$ is the total aboveground biomass, and $w_a$, $w_b$, $w_c$, and $w_d$ are the biomass of wood, bark, branches, and foliage, respectively.

### 2.2.2. Spatial Weight Matrix

The spatial weight matrix ($W$) is a data structure that represents the relationship between locations in geospatial space, and it needs to be considered before constructing a spatial regression model [26]. Distance-based spatial weights and neighborhood-based spatial weights are two major classes of weight functions that can represent the degree of weight or association between different spatial locations in geospatial space [27]. In this study, the spatial weight matrix was constructed using a distance-based bandwidth method. We performed incremental spatial autocorrelation analyses (Moran's I) of biomass for each tree component, with the starting distance set to 5 m and the increment set to 0.5 m. The distance corresponding to the first peak of significance (Z > 1.96) derived from the incremental spatial autocorrelation analyses of biomass was set to bandwidth ($d_s$). This was used to carry out the construction of the spatial weight matrix for the spatial regression model:

$$W_{ij} = \left[ 1 - \left( \frac{d_{ij}}{d_s} \right)^2 \right]^2 \tag{5}$$

where $W_{ij}$ is the spatial weight; $d_s$ is the specified distance, also known as the bandwidth; and $d_{ij}$ is the distance between elements $i$ and $j$.

### 2.2.3. Ordinary Least Squares Model (OLS)

If we want to understand the relationship between a set of n observations on p independent (predictor) variables in $X$ and the dependent (response) variable $Y$, we can perform the following regression with ordinary least squares (OLS) [28]:

$$Y = X\beta + \varepsilon \tag{6}$$

$$\hat{\beta} = \left(X^T X\right)^{-1} X^T Y \tag{7}$$

where $Y$ is a vector of observed response variables, $X$ is an ($n \times k$) model matrix containing one column of 1 (intercept) and p independent variables, $\beta$ is a vector of unknown fixed effect parameters, and $\varepsilon$ is a random error term whose distribution is assumed to obey $N(0, \sigma^2 I)$ [29]. The superscript $T$ denotes the transpose of the matrix; the relationship represented by Equation (3) is universal or constant over the entire geographic region.

### 2.2.4. Spatial Lag Model (SLM)

The SLM is derived from the concept of autoregressive modeling in time series analysis. It is an extension of the classical regression model that takes into account the spatial dependence between data by introducing a spatial lag term [30]. When there is a significant spatial correlation between data and the research that focuses on assessing and predicting this spatial dependence, SLM becomes an appropriate analytical tool [31]. The expression is as follows:

$$Y = X\beta + \lambda W_y + \varepsilon \tag{8}$$

$$Y = (1 - \lambda W)^{-1} X\beta + (1 - \lambda W)^{-1}\varepsilon \tag{9}$$

where $W_y$ represents the spatial lagged response variable, $\lambda$ denotes the spatial autocorrelation coefficient, $W$ is the normalized weight matrix, and $\varepsilon$ is a random error term with distribution $N(0, \sigma^2 I)$. Equation (5) demonstrates that the value of $Y$ at each position is determined by $X$ at that position and by $X$ at adjacent positions through the spatial multiplier $(1-\lambda W)^{-1}$.

### 2.2.5. Spatial Error Model (SEM)

The SEM solves a problem similar to the autocorrelation problem in time series analysis. The SEM is not inherently a purely spatial model. However, it solves the problem of non-independence between the original random error terms by introducing a new spatial lag error term [30]. This spatial lag error term indicates how errors from neighboring regions affect the observations in a given region. Although spatial error does not directly simulate the spatial dependence of the response variable, it indirectly captures spatial effects by considering the spatial dependence of the error term. The expression is as follows:

$$Y = X\beta + \varepsilon = X\beta + \lambda W_\varepsilon + \delta = X\beta + (1 - \lambda W)^{-1}\delta \tag{10}$$

where $W$ is the normalized weight matrix, $W_\varepsilon$ is the spatial error lag term, $\lambda$ is the spatial error term coefficient, and $\delta$ is a well-behaved error term whose distribution is $N(0, \sigma^2 I)$. Equation (6) shows that the response variable at each location is influenced by its explanatory variables and by other location errors through the spatial coefficient $(1-\lambda W)^{-1}$.

### 2.2.6. Spatial Durbin Model (SDM)

The SDM is used to analyze bidirectional spatial dependencies present in spatial data. It combines a spatial lag term and a spatial autoregressive term while taking into account the spatial dependence between the dependent and independent variables [32]. The expression is as follows:

$$Y = \rho W_{1y} + X\beta + \varepsilon \tag{11}$$

$$\varepsilon = \lambda W_{2\varepsilon} + \delta \tag{12}$$

where $W_{1y}$ is the spatial lag counterpart, $\rho$ is the spatial autocorrelation coefficient, $W_{2\varepsilon}$ is the spatial error lag term, and $\lambda$ is the spatial error term coefficient.

### 2.2.7. Geographically Weighted Regression Model (GWR)

The GWR model considers every point in a space [33]. The use of a distance-based weighting function to account for the effects of neighboring points is intended to reveal

patterns of spatial variability by capturing the uniqueness of variable relationships between different locations [34]. This approach emphasizes the heterogeneity of different regions in geospatial space and provides a better understanding of the local characteristics of and trends in spatial data [35]. Assume that each observation $i$ has a set of location coordinates $(u_i, v_i)$. The basic model for geographically weighted regression is

$$Y = \beta_0(u_i, v_i) + \sum_{k}^{n} \beta_k(u_i, v_i)X_k + \delta \tag{13}$$

where $Y$ is the response variable, $(u_i, v_i)$ is the coordinate of the $i$ sample point, $\{\beta_0\, u_i, v_i, \beta_1\, u_i, v_i \ldots \beta_n\, u_i, v_i\}$ is the regression coefficient of the study area location $(u_i, v_i)$, $X_k$ is the explanatory variable, and $\delta$ is a random error term obeying $N(0, \sigma^2\, I)$. GWR is a method for estimating the correlation coefficients of each independent variable $X$ across geographic locations within a specific bandwidth using neighborhood and weighted least squares regression analysis [36]. It is specified as follows:

$$\hat{\beta} = X^T W_i X^{-1} X^T W_i Y \tag{14}$$

where $W_i$ is the geographic weight matrix of center $i$, so $W_i = f(d_i, h)$, $d_i$ is the distance vector between center $i$ and all its neighbors, and $h$ is the distance bandwidth or fading parameter. When $W_i$ is equal to the unit matrix $I$ (constant matrix), this means that each observation has the same weight in the data, at which point the GWR model is effectively equivalent to an OLS model [37].

### 2.3. Model Fitting and Assessment

2.3.1. Model Fitting

The spatial regression model is built by introducing a spatial term based on the optimal base model [30]. In this study, we selected 11 OLS models as base models. The independent variables included DBH, H, CL, and CW, as well as composite variables derived from these factors, while the response variable was the tree component biomass of individual trees in the *Pinus kesiya* var. *langbianensis* forests. The OLS model was fitted using the "nls" package in R. The SLM, SEM, and SDM models were fitted using the "lag", "sem", and "sdm" packages in R, respectively [38]. GWR analysis was performed using the spatial relationship modeling module in ArcGIS [39]. The 11 OLS models are shown below:

$$w_i = a \cdot DBH^b \tag{15}$$

$$w_i = a \cdot DBH^b \cdot H^c \tag{16}$$

$$w_i = a \cdot \left(DBH^2 H\right)^b \tag{17}$$

$$w_i = a \cdot \left(DBH^2 H\right)^b \cdot CW^c \tag{18}$$

$$w_i = a \cdot \left(DBH^2 H\right)^b \cdot CL^c \tag{19}$$

$$w_i = a \cdot DBH^b \cdot H^C \cdot CW^d \tag{20}$$

$$w_i = a \cdot DBH^b \cdot H^C \cdot CL^d \tag{21}$$

$$w_i = a \cdot DBH^b \cdot H^C \cdot CL^d \cdot CW^e \tag{22}$$

$$w_i = a \cdot \left(DBH^2 H\right)^b \cdot CW^c \cdot CL^d \tag{23}$$

$$w_i = a \cdot DBH^b \cdot H^C \cdot \left(CW^2 CL\right)^d \tag{24}$$

$$w_i = a \cdot \left(DBH^2 H\right)^b \cdot \left(CW^2 CL\right)^c \tag{25}$$

where $w_i$ is the biomass of tree components (in kg), *DBH* is the diameter at breast height (in cm), *H* is the tree height (in m), *CL* is the crown length (in m), *CW* is the crown width (in m), and *a*, *b*, *c*, *d*, and *e* are the model parameters.

### 2.3.2. Assessment of OLS Models

The adjusted coefficient of determination ($R_{adj}{}^2$) was chosen as the main metric for assessing model performance [40]. In addition, the Akaike Information Criterion (*AIC*) [41] and Root Mean Square Error (*RMSE*) [42] were used as additional metrics to compare the variability of the models. The criteria for selecting the model were that the *RMSE* values should be close to 0, while the $R_{adj}{}^2$ values should be close to 1, with the smaller *AIC* values indicating higher accuracy. The formulas for these indicators are shown below:

$$R^2_{adj} = 1 - \frac{n - 1 \sum\limits_{i=1}^{n} (y_i - \hat{y}_i)^2}{n - r \sum\limits_{i=1}^{n} (y_i - \overline{y}_i)^2} \tag{26}$$

$$AIC = -2lnL(\hat{\theta}_L, x) + 2q \tag{27}$$

$$RMSE = \sqrt{\frac{\sum(y_i - \hat{y}_i)}{n}} \tag{28}$$

where $y_i$ is the actual observation, $\hat{y}_i$ is the model prediction, $\bar{y}_i$ is the sample mean, $\hat{\theta}_L$ is the maximum likelihood estimate of $\theta$ in the likelihood function ($\hat{\theta}_L$, $x$) of the model, $x$ is the random sample, $q$ is the number of unknown parameters, and $n$ is the number of samples.

### 2.3.3. Assessment of Spatial Regression Models

The spatial regression model was tested to determine how well it applied to the data with the addition of the spatial term compared to the base model. In this study, the SLM, SEM, and SDM models were tested using the likelihood ratio test [30] in R [38]. The correlation and multicollinearity between the local regression coefficients in the GWR model make the traditional likelihood ratio test ineffective [43]. Therefore, we directly compared the AIC values of the GWR model with the AIC values of the OLS model. A difference greater than 2 in AIC was regarded as the GWR model with the addition of the spatial term outperforming the OLS model [44]. This analysis was conducted in ArcMap:

$$\text{LRT} = 2\log\left(\frac{L_2}{L_1}\right) = 2(\log L_2 - \log L_1) \tag{29}$$

where $L_2$ is the great likelihood estimate of the spatial regression models and $L_1$ is the great likelihood estimate of the OLS model.

## 3. Results

### 3.1. Model Fitting

### 3.1.1. OLS Models

We fitted the biomass of the different tree components of individual trees using 11 OLS models. Different models may exist that can more accurately describe changes in tree component biomass due to different ecological environments and growing conditions. Therefore, the optimal model for each tree component was selected based on the model's preference index $R_{adj}{}^2$ (Table 1). This differential selection is intended to provide a more comprehensive understanding of the complexity of tree components in plantation and natural stands and may provide a more accurate basis for subsequent studies, particularly in explaining differences in response to spatial location information when fitting biomass. We observed that the optimal base model varied from one origin to another, from one tree species to another, and from one tree component to another. These differences were mainly due to the inclusion of the crown as an independent variable in the modeling

process, which affected the model-fitting accuracy. In addition, we found that the optimal basic model of the biomass of each tree component for other trees generally has more crown-independent variables than *Pinus kesiya* var. *langbianensis* species in the *Pinus kesiya* var. *langbianensis* forests.

**Table 1.** The optimal base model for biomass estimation is determined for *Pinus kesiya* var. *langbianensis* trees and other trees in the *Pinus kesiya* var. *langbianensis* forest, considering different origins.

| Origin Forms | Tree Components | Models | |
|---|---|---|---|
| | | *Pinus kesiya* **var.** *langbianensis* | Other Trees |
| Plantation | Wood | $w_i = a \cdot (DBH^2H)^b$ | $w_i = a \cdot (DBH^2H)^b \cdot CW^c \cdot CL^d$ |
| | Bark | $w_i = a \cdot DBH^b \cdot H^c$ | $w_i = a \cdot (DBH^2H)^b \cdot (CW^2CL)^c$ |
| | Branches | $w_i = a \cdot DBH^b \cdot H^C \cdot CL^d$ | $w_i = a \cdot DBH^b \cdot H^C \cdot CW^d$ |
| | Foliage | $w_i = a \cdot DBH^b$ | $w_i = a \cdot DBH^b \cdot H^C \cdot CW^d$ |
| | Aboveground | $w_i = a \cdot (DBH^2H)^b \cdot (CW^2CL)^c$ | $w_i = a \cdot (DBH^2H)^b \cdot CW^c \cdot CL^d$ |
| Natural stand | Wood | $w_i = a \cdot (DBH^2H)^b$ | $w_i = a \cdot (DBH^2H)^b \cdot CW^c \cdot CL^d$ |
| | Bark | $w_i = a \cdot DBH^b$ | $w_i = a \cdot (DBH^2H)^b \cdot (CW^2CL)^c$ |
| | Branches | $w_i = a \cdot DBH^b \cdot H^C \cdot (CW^2CL)^d$ | $w_i = a \cdot (DBH^2H)^b \cdot CW^c \cdot CL^d$ |
| | Foliage | $w_i = a \cdot DBH^b \cdot H^C \cdot (CW^2CL)^d$ | $w_i = a \cdot DBH^b \cdot H^C \cdot CW^d$ |
| | Aboveground | $w_i = a \cdot (DBH^2H)^b \cdot (CW^2CL)^c$ | $w_i = a \cdot DBH^b \cdot H^C \cdot CW^d$ |

Note: $w_i$ is the biomass of tree components, the unit is kg; *DBH* is the diameter at breast height, the unit is cm; *H* is the tree height, the unit is m; *CL* is the crown length, the unit is m; *CW* is the crown width, the unit is m; *a*, *b*, *c*, *d*, are the model parameters.

3.1.2. Spatial Regression Models

The results of model fitting for each tree component are reported in Tables 2 and 3. After introducing the spatial location information to the modeling process, the fitting effects of the tree components changed. The fitting accuracy of the total biomass for branches, foliage, and aboveground parts of trees in the plantation was significantly improved. That is, the $R_{adj}^2$ increased by 0.08 and 0.17, AIC decreased by 26.2 and 28.48, and RMSE reduced was by 0.51 and 0.41 for *Pinus kesiya* var. *langbianensis* and other tree species' branches, respectively. For foliage, the $R_{adj}^2$ increased by 0.1 and 0.17, AIC decreased by 21.32 and 15.7, and RMSE was reduced by 0.26 and 0.16 for *Pinus kesiya* var. *langbianensis* and the other tree species, respectively. The $R_{adj}^2$ of aboveground biomass increased by 0.03 and 0.04, AIC decreased by 13.93 and 5.97, and RMSE was reduced by 1.43 and 0.98.

The fitting performance was improved for all tree components of trees in the natural stand. For wood, the $R_{adj}^2$ increased by 0.01 and 0.03, AIC decreased by 3.47 and 148.74, and RMSE was reduced by 4.93 and 4.06 for *Pinus kesiya* var. *langbianensis* and the other tree species, respectively. For bark, the $R_{adj}^2$ increased by 0.13 and 0.26, AIC decreased by 12.89 and 328.25, and RMSE was reduced by 0.42 and 3.61. For branches, the $R_{adj}^2$ increased by 0.05 and 0.24, the AIC decreased by 12.79 and 155.48, and the RMSE was reduced by 1.89 and 2.77. Finally, for foliage, the $R_{adj}^2$ increased by 0.1 and 0.03, AIC decreased by 17.27 and 9.06, and RMSE was reduced by 0.23 and 0.07. This suggests that the fitting accuracy of the crown parts of the tree in the plantation and all tree components above ground in the natural stand was mostly affected by the spatial position.

**Table 2.** Model evaluation indicators for biomass of various tree components of *Pinus kesiya* var. *langbianensis* trees in plantation and natural stand.

| Tree Components | Models | Plantation | | | | | Natural Stand | | | | |
|---|---|---|---|---|---|---|---|---|---|---|---|
| | | $R_{adj}^2$ | AIC | RMSE | LRT-Value | *p*-Value | $R_{adj}^2$ | AIC | RMSE | LRT-Value | *p*-Value |
| Wood | OLS | 0.80 | 2084.65 | 12.78 | - | - | 0.88 | 1533.24 | 78.73 | - | - |
| | SLM | 0.80 | 2087.00 | 12.79 | 0.31 | 0.58 | 0.88 | 1534.13 | 78.40 | 0.52 | 0.47 |
| | SEM | 0.80 | 2087.28 | 12.80 | 0.04 | 0.85 | 0.88 | 1531.78 | 76.87 | 2.87 | 0.09 |
| | SDM | 0.80 | 2088.01 | 12.76 | 0.52 | 0.47 | 0.89 | 1529.79 | 75.01 | 5.64 | 0.02 |
| | GWR [1][2] | 0.80 | 2084.44 | 12.70 | - | - | 0.89 | 1529.77 | 73.80 | - | - |
| Bark | OLS | 0.47 | 1517.95 | 4.32 | - | - | 0.15 | 800.40 | 4.90 | - | - |
| | SLM [1] | 0.46 | 1520.51 | 4.32 | 0.13 | 0.72 | 0.14 | 806.36 | 4.90 | 0.30 | 0.58 |
| | SEM | 0.46 | 1520.81 | 4.32 | 0.38 | 0.54 | 0.16 | 804.57 | 4.84 | 2.09 | 0.15 |
| | SDM [2] | 0.46 | 1524.85 | 4.32 | 0.28 | 0.60 | 0.28 | 787.51 | 4.48 | 9.24 | 0.00 |
| | GWR | 0.46 | 1519.35 | 4.35 | - | - | 0.15 | 800.90 | 4.95 | - | - |
| Branches | OLS | 0.30 | 1858.92 | 8.24 | - | - | 0.59 | 1283.05 | 30.87 | - | - |
| | SLM | 0.37 | 1838.49 | 7.83 | 25.24 | 0.00 | 0.59 | 1285.47 | 30.09 | 0.87 | 0.35 |
| | SEM [1] | 0.38 | 1832.72 | 7.73 | 30.53 | 0.00 | 0.59 | 1286.37 | 30.19 | 0.28 | 0.60 |
| | SDM | 0.37 | 1834.03 | 7.83 | 0.03 | 0.87 | 0.62 | 1285.69 | 28.97 | 4.30 | 0.04 |
| | GWR [2] | 0.36 | 1838.35 | 7.69 | - | - | 0.64 | 1270.26 | 28.98 | - | - |
| Foliage | OLS | 0.15 | 1528.07 | 4.42 | - | - | 0.41 | 722.55 | 3.60 | - | - |
| | SLM | 0.23 | 1508.69 | 4.21 | 22.55 | 0.00 | 0.44 | 719.09 | 3.51 | 2.01 | 0.16 |
| | SEM | 0.23 | 1508.38 | 4.20 | 22.86 | 0.00 | 0.44 | 719.99 | 3.52 | 1.12 | 0.29 |
| | SDM | 0.20 | 1517.53 | 4.27 | 14.62 | 0.00 | 0.49 | 718.07 | 3.35 | 6.07 | 0.01 |
| | GWR [1][2] | 0.25 | 1506.75 | 4.16 | - | - | 0.51 | 705.28 | 3.37 | - | - |
| Aboveground | OLS | 0.78 | 2306.94 | 19.46 | - | - | 0.88 | 1580.01 | 93.29 | - | - |
| | SLM | 0.78 | 2305.32 | 19.24 | 2.52 | 0.11 | 0.88 | 1580.74 | 92.82 | 1.41 | 0.24 |
| | SEM | 0.80 | 2292.13 | 18.65 | 15.71 | 0.00 | 0.88 | 1581.38 | 92.84 | 0.76 | 0.38 |
| | SDM | 0.80 | 2287.31 | 18.36 | 1.07 | 0.30 | 0.89 | 1577.14 | 89.37 | 4.62 | 0.03 |
| | GWR [1][2] | 0.81 | 2293.01 | 18.03 | - | - | 0.93 | 1580.58 | 70.50 | - | - |

Note: The optimal spatial regression model for plantation is represented by superscript 1, and the optimal spatial regression model for natural stand is represented by superscript 2.

**Table 3.** Model evaluation indicators for biomass of various tree components of other trees in plantation and natural stand.

| Tree Components | Models | Plantation | | | | | Natural Stand | | | | |
|---|---|---|---|---|---|---|---|---|---|---|---|
| | | $R_{adj}^2$ | AIC | RMSE | LRT-Value | *p*-Value | $R_{adj}^2$ | AIC | RMSE | LRT-Value | *p*-Value |
| Wood | OLS | 0.87 | 524.93 | 3.54 | - | - | 0.91 | 2583.22 | 16.34 | - | - |
| | SLM | 0.86 | 539.69 | 3.78 | 0.00 | 0.97 | 0.90 | 2582.92 | 16.37 | 0.96 | 0.33 |
| | SEM | 0.87 | 535.34 | 3.67 | 4.36 | 0.04 | 0.84 | 2721.02 | 20.53 | 0.12 | 0.73 |
| | SDM | 0.87 | 538.70 | 3.64 | 0.28 | 0.60 | 0.84 | 2720.62 | 20.32 | 0.22 | 0.64 |
| | GWR [1][2] | 0.89 | 520.24 | 3.04 | - | - | 0.94 | 2434.48 | 12.28 | - | - |
| Bark | OLS | 0.88 | 245.83 | 0.84 | - | - | 0.62 | 2170.79 | 8.36 | - | - |
| | SLM | 0.87 | 251.01 | 0.85 | 2.35 | 0.13 | 0.63 | 2167.13 | 8.31 | 0.04 | 0.85 |
| | SEM | 0.87 | 253.36 | 0.86 | 0.00 | 0.95 | 0.63 | 2165.77 | 8.28 | 1.39 | 0.24 |
| | SDM | 0.88 | 248.39 | 0.81 | 2.71 | 0.10 | 0.63 | 2166.42 | 8.24 | 0.06 | 0.80 |
| | GWR [1][2] | 0.90 | 239.26 | 0.76 | - | - | 0.88 | 1842.54 | 4.75 | - | - |
| Branches | OLS | 0.48 | 443.13 | 2.31 | - | - | 0.58 | 2153.61 | 8.13 | - | - |
| | SLM | 0.49 | 443.68 | 2.29 | 1.41 | 0.24 | 0.58 | 2155.64 | 8.12 | 1.34 | 0.25 |
| | SEM | 0.49 | 443.31 | 2.27 | 1.78 | 0.18 | 0.58 | 2156.19 | 8.13 | 0.79 | 0.37 |
| | SDM [1] | 0.65 | 414.65 | 1.90 | 1.22 | 0.27 | 0.59 | 2150.20 | 8.07 | 0.06 | 0.81 |
| | GWR [2] | 0.61 | 428.74 | 2.02 | - | - | 0.82 | 1998.13 | 5.36 | - | - |

**Table 3.** *Cont.*

| Tree Components | Models | Plantation | | | | | Natural Stand | | | | |
|---|---|---|---|---|---|---|---|---|---|---|---|
| | | $R_{adj}^2$ | AIC | RMSE | LRT-Value | *p*-Value | $R_{adj}^2$ | AIC | RMSE | LRT-Value | *p*-Value |
| Foliage | OLS | 0.25 | 327.92 | 1.27 | - | - | 0.36 | 1498.20 | 2.78 | - | - |
| | SLM [2] | 0.26 | 328.10 | 1.26 | 0.03 | 0.86 | 0.39 | 1489.14 | 2.71 | 5.52 | 0.02 |
| | SEM | 0.26 | 328.09 | 1.26 | 0.04 | 0.84 | 0.37 | 1494.65 | 2.75 | 0.00 | 0.95 |
| | SDM [1] | 0.42 | 312.22 | 1.11 | 2.34 | 0.13 | 0.39 | 1492.79 | 2.70 | 2.69 | 0.10 |
| | GWR | 0.36 | 323.81 | 1.20 | - | - | 0.38 | 1490.83 | 2.74 | - | - |
| Aboveground | OLS | 0.89 | 610.69 | 5.03 | - | - | 0.90 | 2803.56 | 23.59 | - | - |
| | SLM | 0.86 | 616.79 | 5.64 | 0.66 | 0.42 | 0.90 | 2812.60 | 23.92 | 5.31 | 0.02 |
| | SEM | 0.87 | 610.53 | 5.41 | 6.91 | 0.01 | 0.89 | 2817.90 | 24.15 | 0.02 | 0.89 |
| | SDM | 0.89 | 605.19 | 5.12 | 2.46 | 0.12 | 0.90 | 2815.11 | 23.86 | 0.79 | 0.37 |
| | GWR [1,2] | 0.93 | 604.72 | 4.05 | - | - | 0.91 | 2806.21 | 22.10 | - | - |

Note: The optimal spatial regression model for plantation is represented by superscript 1, and the optimal spatial regression model for natural stand is represented by superscript 2.

### 3.2. Differences in the Response of Aboveground Tree Components to Spatial Location

We tested the spatial terms in the construction of the spatial regression model (Tables 2 and 3). When the spatial regression model was used to fit the biomass of different tree components, adding spatial terms created significant differences. The spatial terms had significant effects ($p < 0.05$ or $\Delta AIC > 2$) on the biomass of all tree components except the wood and bark portion of the *Pinus kesiya* var. *langbianensis* in the plantation (Table 4). This suggests that the spatial–positional relationships between individual trees of the *Pinus kesiya* var. *langbianensis* in the plantation are controlled by stand origin. In predicting the biomass of wood and bark fractions, traditional OLS models are sufficiently explanatory. In contrast, for the estimation of branch, foliage, and aboveground biomass of *Pinus kesiya* var. *langbianensis* in the plantation and biomass of all aboveground tree components of the other tree species in the plantation and all trees in the natural stand, we need to take into account different spatial–positional relationship differences between different tree components. Therefore, spatial regression modeling is a better choice in this case.

**Table 4.** The $\Delta R_{adj}^2$, $\Delta RMSE$, and $\Delta AIC$ of the optimal spatial regression model and base model.

| Origins | Tree Species | Index | Different Tree Components | | | | | Total |
|---|---|---|---|---|---|---|---|---|
| | | | Wood | Bark | Branches | Foliage | Aboveground | |
| Plantation | *Pinus kesiya* var. *langbianensis;* | $\Delta R_{adj}^2$ | 0.00 | −0.01 | 0.08 | 0.10 | 0.03 | 0.20 |
| | | $\Delta RMSE$ | −0.08 | 0.00 | −0.51 | −0.26 | −1.43 | −2.28 |
| | | $\Delta AIC$ | −0.21 | 2.56 | −26.2 | −21.32 | −13.93 | −59.10 |
| | Other trees | $\Delta R_{adj}^2$ | 0.02 | 0.02 | 0.17 | 0.17 | 0.04 | 0.42 |
| | | $\Delta RMSE$ | −4.93 | −0.42 | −1.89 | −0.23 | −22.79 | −30.26 |
| | | $\Delta AIC$ | −4.69 | −6.57 | −28.48 | −15.70 | −5.97 | −61.41 |
| Natural stand | *Pinus kesiya* var. *langbianensis;* | $\Delta R_{adj}^2$ | 0.01 | 0.13 | 0.05 | 0.10 | 0.05 | 0.34 |
| | | $\Delta RMSE$ | −0.50 | −0.08 | −0.41 | −0.16 | −0.98 | −2.13 |
| | | $\Delta AIC$ | −3.47 | −12.89 | −12.79 | −17.27 | 0.57 | −45.85 |
| | Other trees | $\Delta R_{adj}^2$ | 0.03 | 0.26 | 0.24 | 0.03 | 0.01 | 0.57 |
| | | $\Delta RMSE$ | −4.06 | −3.61 | −2.77 | −0.07 | −1.49 | −12.00 |
| | | $\Delta AIC$ | −148.74 | −328.25 | −155.48 | −9.06 | 2.65 | −638.88 |

Note: $\Delta R_{adj}^2$, $\Delta RMSE$, and $\Delta AIC$ were obtained by subtracting the $R_{adj}^2$, RMSE, and AIC of the OLS from the $R_{adj}^2$, RMSE, and AIC of optimal spatial regression model.

### 3.3. Effects of Diameter at Breast Height on the Response of Tree Components to Spatial Location

We compared the differences between the different DBHs to take into account—as well as not take into account—spatial location information (Figures 4–7). According to

the results, the differences in the predicted values for the crown at different DBHs were significantly greater than those for the wood and bark. It can be seen from Figures 4–7 that for the crown, the area enclosed by the OLS model and the optimal spatial regression model is significant starting from DBH = 5 cm. On the other hand, when using a model that considers spatial position relationships to fit crown biomass, the $R_{adj}^2$ of *Pinus kesiya* var. *langbianensis* species and other trees increased by 0.18, 0.34, and 0.15, 0.27 in the plantation and natural stands, respectively (Table 4). This shows that there are already differences in the impact of considering and not considering spatial position relationships on the crown biomass starting from DBH = 5 cm. The predicted values of the spatial regression model were significantly larger than the predicted values of the OLS model. As the DBH increased, the fitted differences between the branch and foliage biomass of *Pinus kesiya* var. *langbianensis* became insignificant at DBH = 19.5 cm and DBH = 20.3 cm, and the branch biomass of the other tree species became insignificant at DBH = 14 cm in the plantation. The branch and foliage biomass of *Pinus kesiya* var. *langbianensis* changed insignificantly at DBH = 34.6 cm and DBH = 31.4 cm, and the branch biomass of the other tree species changed insignificantly at DBH = 19.5 cm in the natural stand. The aboveground biomass of the other tree species changed insignificantly at DBH = 19.5 cm in the natural stand. The differences became significant again with continued growth in DBH but with an opposite trend to the earlier changes. This suggests that these DBH sizes are thresholds that affect the response of the biomass of each tree component to spatial effects.

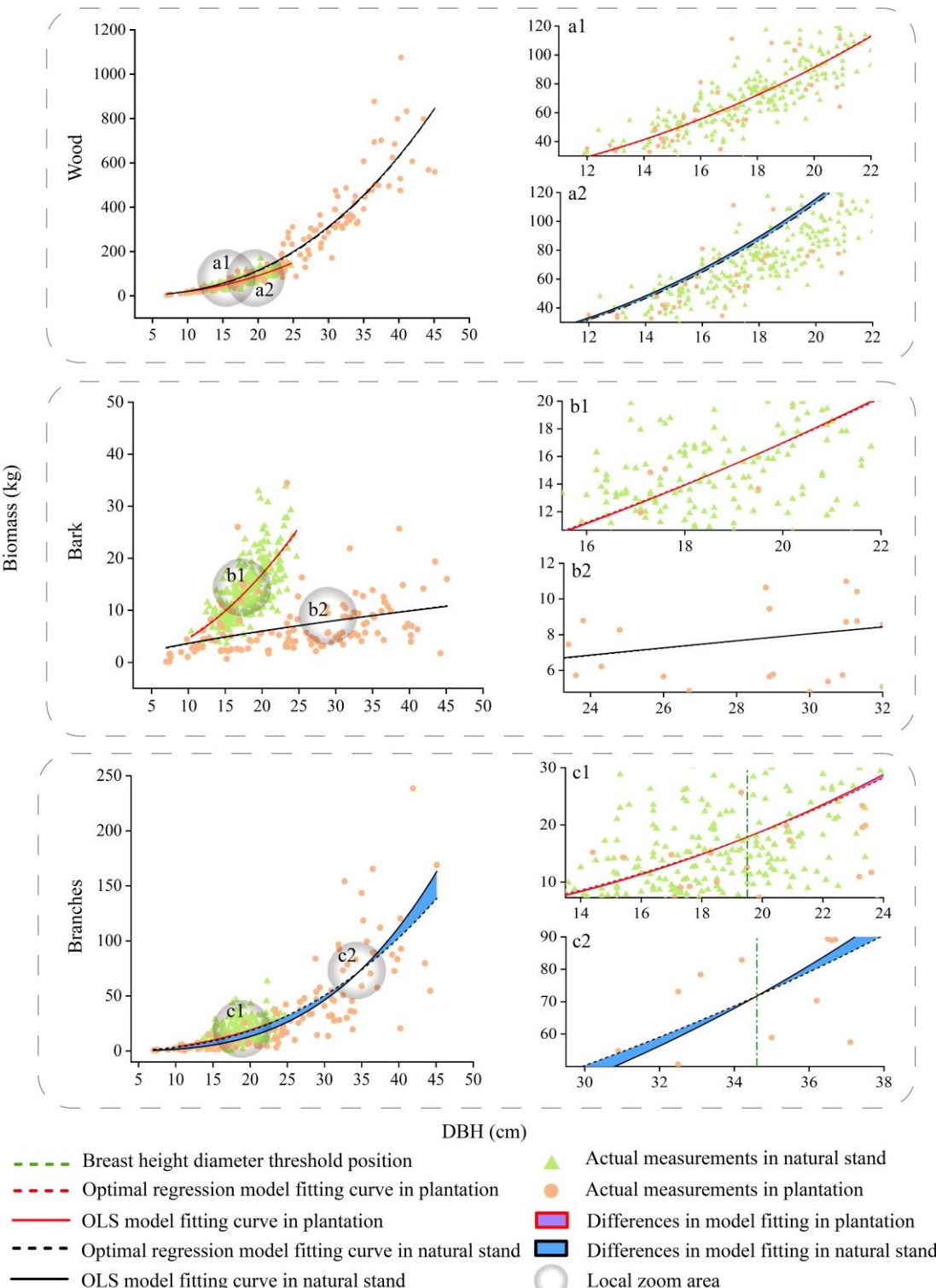

**Figure 4.** Fitting curves of the basic model and optimal spatial regression model for the wood, bark, and branches biomass of *Pinus kesiya* var. *langbianensis* trees. a1 and a2 are localised enlargements of wood from plantation and natural stand. b1 and b2 are localised enlargements of bark from plantation and natural stand. c1 and c2 are localised enlargements of branches from plantation and natural stand.

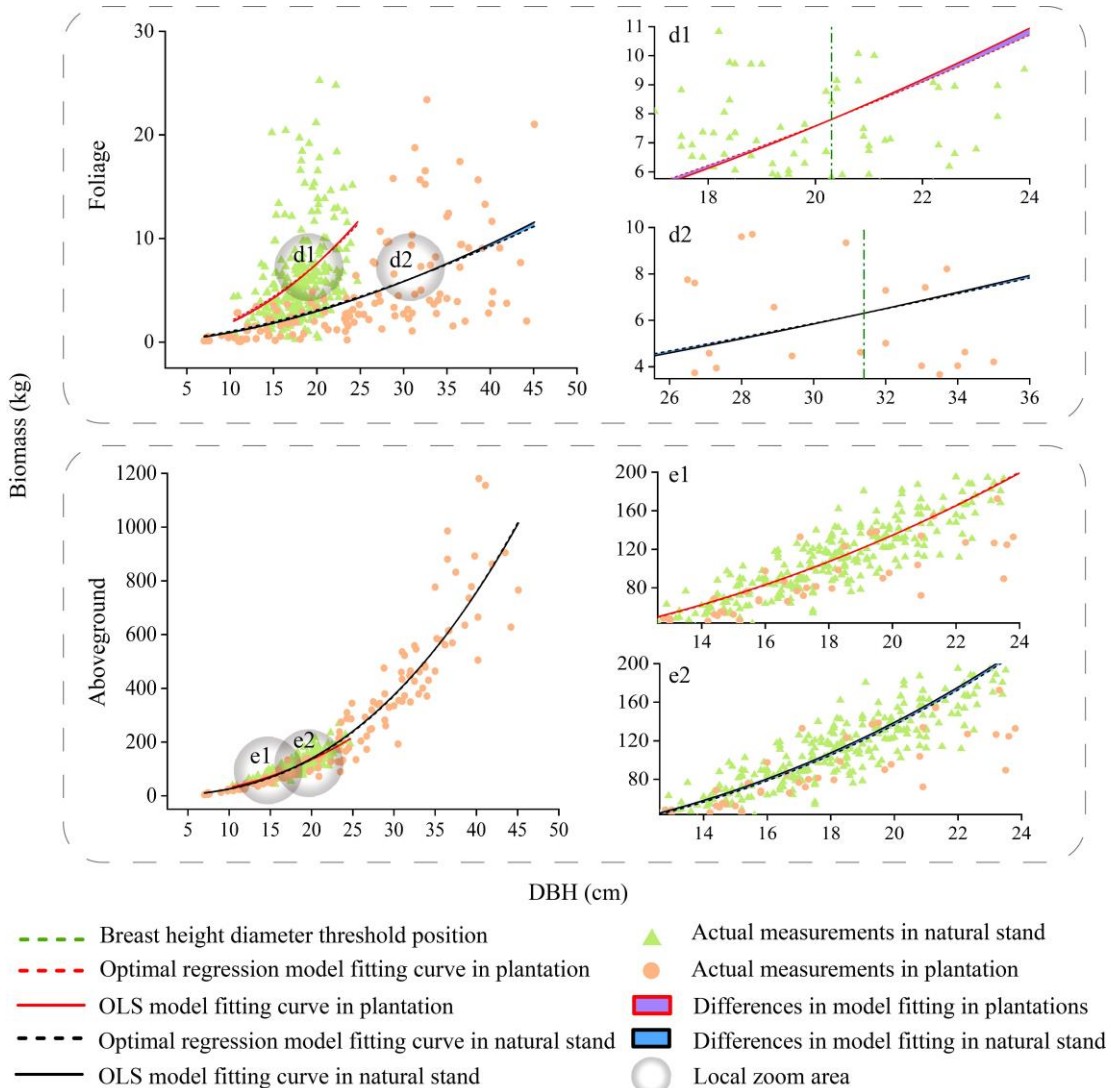

**Figure 5.** Fitting curves of the basic model and optimal spatial regression model for the foliage, and aboveground biomass of *Pinus kesiya* var. *langbianensis* trees. d1 and d2 are localised enlargements of foliage from plantation and natural stand. e1 and e2 are localised enlargements of aboveground biomass from plantation and natural stand.

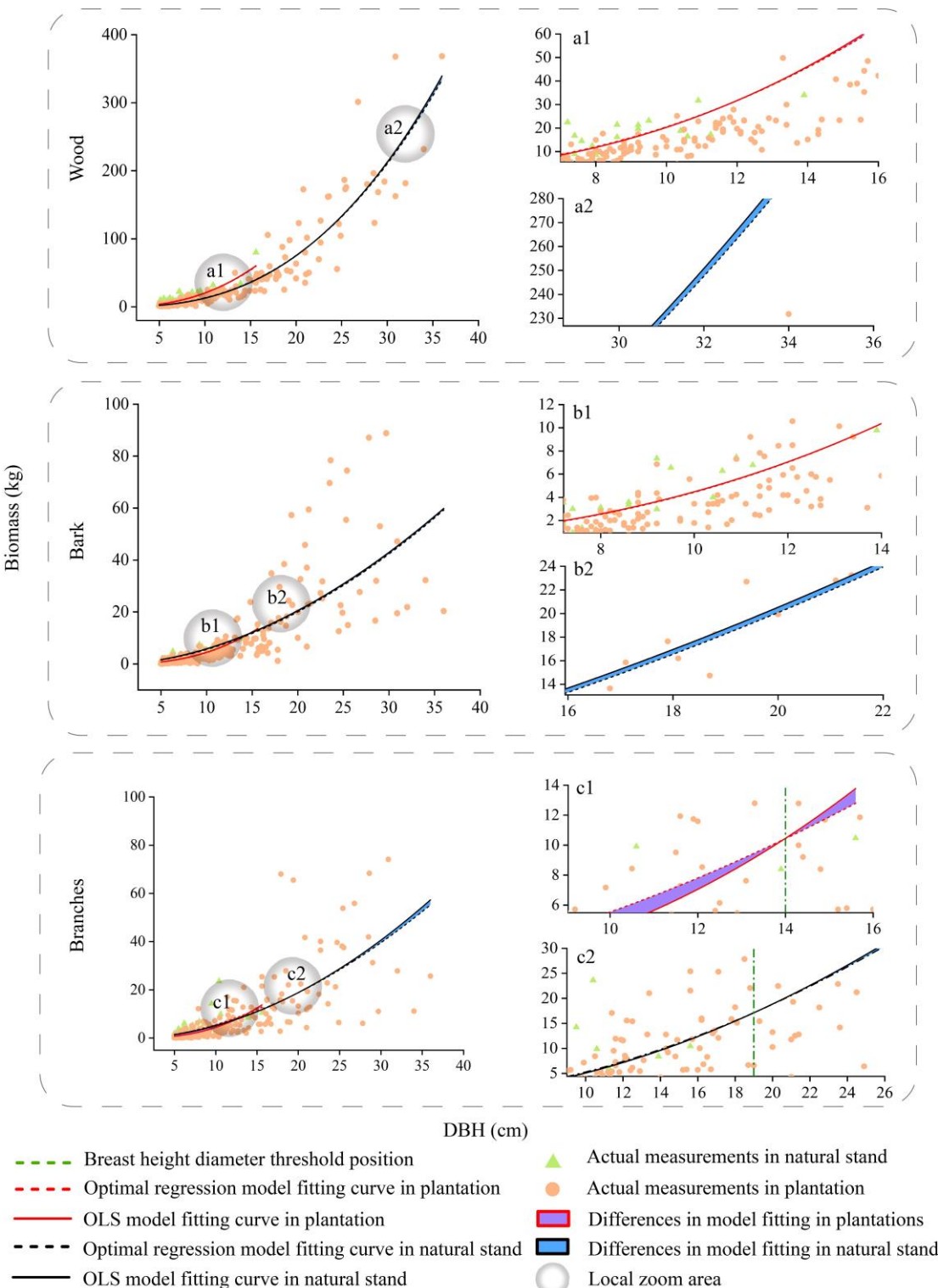

**Figure 6.** Fitting curves of the basic model and optimal spatial regression model for the wood, bark, and branch biomass of other trees. a1 and a2 are localised enlargements of wood from plantation and natural stand. b1 and b2 are localised enlargements of bark from plantation and natural stand. c1 and c2 are localised enlargements of branches from plantation and natural stand.

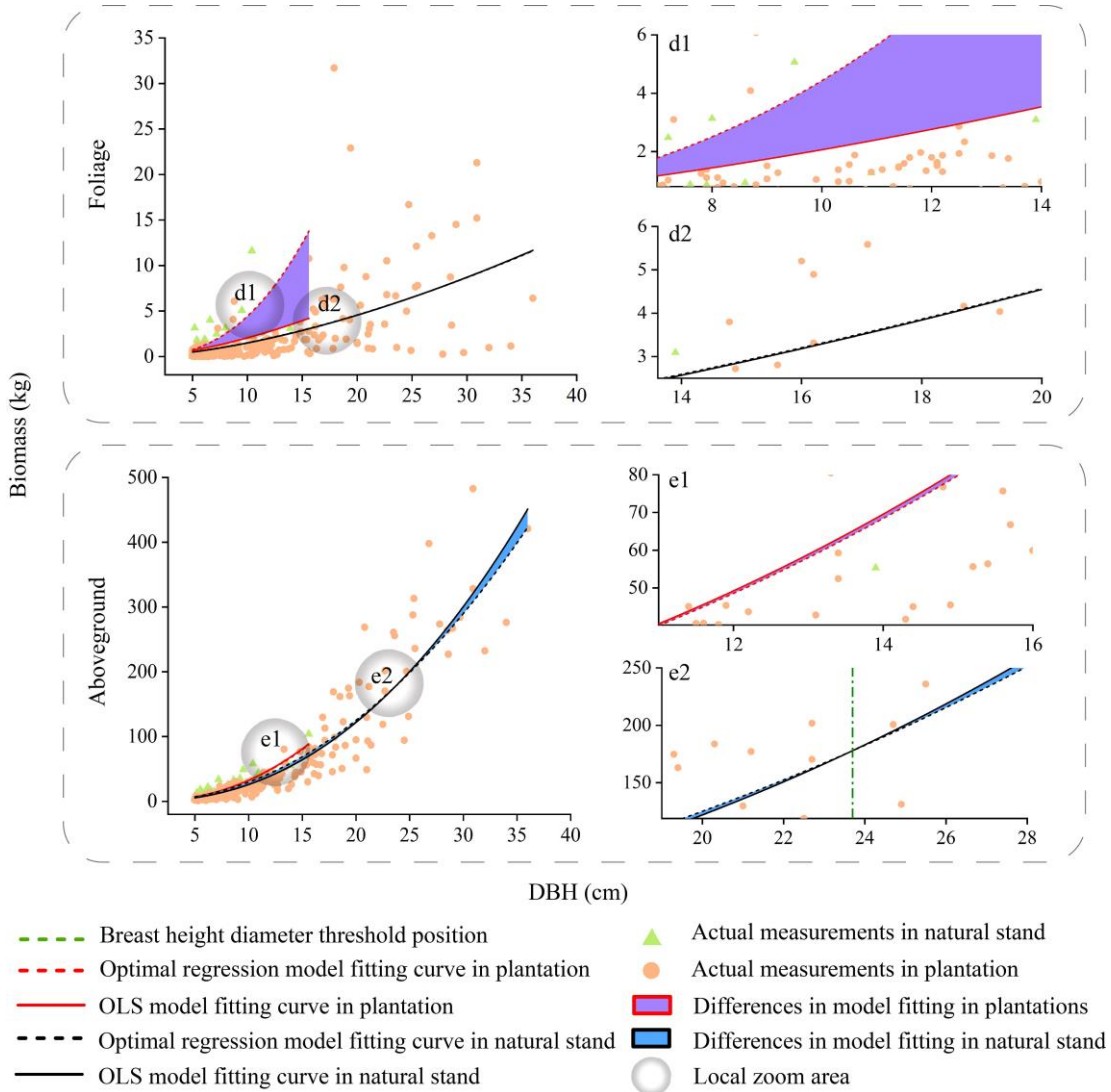

**Figure 7.** Fitting curves of the basic model and optimal spatial regression model for the foliage and aboveground biomass of other trees. d1 and d2 are localised enlargements of foliage from plantation and natural stand. e1 and e2 are localised enlargements of aboveground biomass from plantation and natural stand.

## 4. Discussion

### 4.1. Choice of Methodology and Application of Data

#### 4.1.1. Spatial Regression Model

We fitted the biomass of each tree component (wood, bark, branches, foliage) and the total aboveground biomass of individual trees from different origins (plantation and natural stand) using methods that either took spatial location information into account (SLM, SEM, SDM, and GWR models) or did not (OLS model). The fitting accuracy of the spatial regression model was better than that of the OLS model. This was because the spatial regression model could deal with the spatial dependence between the observation and the spatial locations to reflect differences in the response of tree growth to these spatial locations [30]. In addition, the spatial term significantly indicated that the OLS (underlying model) no longer met the requirements for the independent and dependent variables that existed in the spatial location relationship [26]. Therefore, it was meaningful to apply spatial regression modeling to analyze the effect of spatial location on tree growth [37].

#### 4.1.2. Selection of Spatial Weight Matrix

Selecting the spatial weights is crucial for constructing spatial regression models, and distance-based spatial weights and neighbor-based spatial weights are common methods for constructing spatial weight matrices [27]. In this study, the spatial weight matrix was constructed based on the distance between trees. Neighbor-based spatial weights need to consider differences in tree species and the existence of promotive, epiphytic, or inhibitory relationships [45]. The botanical heterogeneity of a forest means that a tree and its neighbor may be of the same or different species [46], and whether their interactions are spatially positive, negative, or irrelevant is currently unknown. In contrast, distance-based spatial weights consider the distance of spatial affecting each other between trees, as whether or not trees are neighbors cannot explain the consistency in spatial interactions between trees [47]. Therefore, the distance-based spatial weights were more accurate for representing the spatial influence between trees in this study.

#### 4.1.3. Availability of Data

We used the fitted variance in tree biomass to reveal the spatial diversity of tree growth because this can reflect the process and state of tree growth [48]. Exploring differences in tree growth responses to spatial location requires other factors to be the same or similar [49]. Therefore, we selected both plantation and natural stands located in the subtropical monsoon zone, with similar stand conditions and stand structures. Climatic conditions and soil types in which trees grew did not vary greatly within the stands, suggesting that the pattern of tree growth change was caused by spatial effects rather than other factors. Time produced negligible differences in tree growth in our study since the trees were cut at a time that did not span one cycle of their growth and were cut in the same manner [50]. In addition, the majority of previous studies' data used in spatial effects research were based on large-scale forest inventories or remote sensing, which led to a lower precision regarding spatial effects between individual trees [37,51,52]. However, we used the sample-cutting data at the same stage as more precisely recorded data (DBH, H, CL, CW) to reveal individual tree growth patterns. The data were more accurate for expressing the distribution pattern of biomass of an individual tree by tree component [53]. On the other hand, the clear-cut sample plots selected in this article are similar to other local forest communities of the same origin in terms of geographical location, ecological environment, and vegetation type. This choice ensures that our study is somewhat representative of the entire local forest ecosystem. For example, Dung and Kim [54] studied runoff and soil erosion during the clear-cutting period of acacia plantations in the headwater mountains of Vietnam based on one experimental plot and one control plot. And Vestin et al. [55] based on two clear-cut plots with different humidity, the impact of clear-cutting on greenhouse gas flux was studied, and the impact of clear-cut plots with different humidity was compared. The datasets they used were obtained from a representative plot that was capable of representing the characteristics of the local forest type. Therefore, our research results have a certain reference value for understanding the overall status of local forest communities.

### 4.2. Response of Tree Biomass to Different Spatial Locations
#### 4.2.1. Impact of Origins

We found a stronger degree of responsiveness of individual tree biomass to spatial–positional relationships within the natural stand than in the plantation. The $\Delta R_{adj}^2$ increase range for tree components in the plantation was $-0.1$ to $0.17$, totaling $0.62$. The $\Delta$RMSE decrease ranged from $0.00$ to $22.79$, totaling $32.51$, while the $\Delta$AIC decrease ranged from $-2.56$ to $28.48$, totaling $120.51$. In the natural stand, the $\Delta R_{adj}^2$ increase ranged from $0.1$ to $0.26$, totaling $0.91$. The $\Delta$RMSE decrease range ranged from $0.07$ to $4.06$, totaling $14.13$, and the $\Delta$AIC decrease range ranged from $-2.65$ to $328.25$, totaling $684.73$ (Table 4). Natural stands are spontaneously formed forest stands that are affected by environmental factors such as climate, soil, and topography, which have richer species diversity per unit area and more complex spatial patterns [56]. Thus, the growth space of individual standing

trees may be affected by an increase in the number and size of surrounding trees, resulting in rapid responsiveness in terms of resource utilization and growth strategies [57,58]. On the other hand, plantations already control the size of spatial positions between trees at the early stage of forestation, leading to lower distance heterogeneity [59]. Moreover, the spatial–positional relationships are always maintained at a level most favorable for tree growth due to human control of certain factors in the management of the forest. As a result, the ecological characteristics and biomass of individual *Pinus kesiya* var. *langbianensis* trees within the natural stand in our study were more strongly influenced by spatial effects than those in the plantation. However, the foliage of the other tree species showed the opposite trend (Figure 7). This may be because the other tree species had grown under the growth space of *Pinus kesiya* var. *langbianensis* in the plantation, while *Pinus kesiya* var. *langbianensis* and the other tree species had relatively independent growth spaces in the natural stand, as illustrated in Figure 2. As a result, the foliage of the other tree species responded to spatial effects to a greater extent in the plantation than in the natural stand.

### 4.2.2. Differences in the Effects of DBH Size

Further analyses showed DBH was also a factor in the variance in responses of biomass to the spatial location relationship. The OLS model underestimated the individual tree biomass before the DBH threshold and overestimated it after the threshold (Figures 4–7). This may be because the growth rates [60], competitive relationships [61], and nutrient utilization [62] had changed along with the increasing tree size. At early stages of growth (from DBH = 5 cm), the effect of the spatial location relationship is positive, as it reduces the constraints on tree growth imposed by environmental factors (e.g., provision of partial shade, resource sharing, wind-blocking effects, and competitive incentives) [63]. When a tree reaches the DBH competition threshold, the tree enters an intermediate period between symmetric and asymmetric competition, and the effect of spatial–positional relationships on tree growth is smallest [23,64]. As spatial effects begin to constrain tree growth with increasing DBH, trees that occupy the dominant ecological niches may gain more growth space via competition and simultaneously constrain the growth of the remaining trees, leading to their elimination [65]. At this point, the spatial–positional relationship is negative to tree growth. Therefore, we need to be aware of the DBH competition threshold caused by the spatial location relationship, which affects the process of forest management.

As shown in Figures 4–7, the competition threshold of DBH was reached at 19.5 cm and 34.6 cm for branches and 20.3 cm and 31.4 cm for foliage in *Pinus kesiya* var. *langbianensis* in the plantation and natural stand, respectively. Only the branches had an obvious competition threshold for the other tree species, which was 14 cm and 19 cm in the plantation and natural stand, respectively. In natural stands, the diversity of species necessitates the utilization of the vertical space to reduce spatial competition and weaken intraspecific competition [66]. The efficient layered structure causes the lower vegetation to take longer to achieve asymmetric competition for growth space with the upper vegetation [67]. In our study, the spatial competition relationship of trees in the natural stand was generally symmetric to asymmetric, and this competition transition always occurred in trees with greater DBH. In contrast, the heterogeneity of DBH and H was lower in the plantation (Figure 3), and the vertical space could not be utilized efficiently. The plantation stand canopy closed when DBH was relatively small compared to the same stocking in one unit area of the natural stand, with horizontal spatial competition for the dominant niche for growth resources occurring [68]. Therefore, trees in the plantation had a smaller DBH when they reached the competition threshold than those in the natural stand.

### 4.2.3. Reactions of Different Tree Components

Some differences were also observed in the responses of the individual tree components to their spatial–positional relationships (Figures 4–7). The fitting accuracy of the crown was further improved under consideration of the spatial–positional relationship. Competition between trees and interactions between ecological environments lead to a

difference in resource allocation in the trunks, crowns, and roots [69,70]. Hence, the impact of spatial–positional relationships on the growth of different tree components varies. The distance between trees is generally measured based on the position of the trunk. However, the growth space of the trunk may not be the standard for other tree components due to differences in their morphological characteristics and growth space requirements [71,72]. The field of influence of the spatial–positional relationship was narrower for the crown than for the trunk. While the trunk growth of an individual tree is usually retained within its own growth space, its crown may encroach on the growth space of neighboring trees [46]. This encroachment may involve the crown or the trunk of the adjacent tree, largely based on its size [73,74]. Accordingly, this illustrates that differences in spatial effects of trunks and crowns on surrounding trees need to be addressed when estimating stand biomass. This is a positive contribution to our understanding of the characteristics of resource allocation within individual trees and gives an in-depth insight into the functions and dynamic processes of forest ecosystems.

### 4.3. Limitations and Future Research

This study focused on the aboveground tree components of individual trees; however, roots also play a key role in the total biomass of individual trees and belowground carbon storage [75]. Therefore, future research should consider the root biomass to clarify the spatial–positional relationship of the entire individual tree. In addition, whether the spatial location or environmental factors are the dominant drivers of competition or facilitation between two neighboring (different) tree species needs to be explored. Moreover, the spatial impact of trees with DBH < 5 cm and understory vegetation is unclear and needs to be researched in the future.

### 5. Conclusions

In the present work, we used OLS models and spatial regression models (SLM, SEM, SDM, and GWR) to fit the biomass of each tree component for individual trees in *Pinus kesiya* var. *langbianensis* forests of different tree components. This was used to compare differences in biomass fitting accuracy that were affected by spatial effects of the different origins, tree components, and DBH. Our study shows that there was a stronger influence of spatial effects on the fitting accuracy within the natural stand and that the fitting accuracy of aboveground tree components was improved by using the spatial regression models. The spatial effects within the plantation mainly influenced the crown (branches and foliage) biomass, and the fitting accuracy was significantly improved by using the spatial regression models. Spatial effects had a positive effect on trees with small DBH within the stand. The spatial effects were minimal when tree DBH was at an intermediate stage between size-symmetric and -asymmetric competition. Intense competition for spatial location occurred between large DBH trees and their neighboring trees that exceeded the DBH threshold. It is worth noting that trees had a smaller DBH when they reached the competition threshold in the plantation than in the natural stand. At this point, the spatial effects began to restrict tree growth. Further analyses showed that spatial relationships were more pronounced in changing from positive to negative effects on the crown than trunk (wood and bark) growth. Therefore, we suggest controlling the spatial–positional relationship of tree components according to DBH in forest management to enhance the carbon stock of forests and achieve better ecological benefits. This work is a useful reference for high-precision forest biomass estimation and management objectives oriented toward carbon stock enhancement.

**Author Contributions:** C.L.: Participated in the collection of the data, conducted the data analysis, and wrote the draft of the paper; Y.W., X.Z., H.L., Z.Y., Z.L., W.L. and Q.F.: Helped with the data analysis and some of the graphs; G.O.: Supervised and coordinated the research project, designed the experiment, and revised the paper. All authors have read and agreed to the published version of the manuscript.

**Funding:** This research was jointly supported by the Key Research and Development Program of Yunnan Province, China (202303AC100009), and the Ten Thousand Talent Plans for Young Top-notch Talents of Yunnan Province (YNWR-QNBJ-2018-184).

**Data Availability Statement:** The original contributions presented in the study are included in the article, further inquiries can be directed to the corresponding authors.

**Acknowledgments:** We would like to acknowledge all the people who have contributed to this paper.

**Conflicts of Interest:** The authors of this manuscript declare that they have no financial or non-financial conflicts of interest that could potentially influence the research reported in this paper. Specifically, the authors declare that: They have not received any financial support or funding that could bias their interpretation or reporting of the research findings. They have no personal relationships with any individuals or organizations that could influence their research. They have not engaged in any activities that could create a conflict of interest, such as consulting work or expert testimony for companies or organizations related to the research. In addition, the authors confirm that the research reported in this paper is original and has not been previously published or is under consideration for publication elsewhere.

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
