# Peer review of "Response of Individual-Tree Aboveground Biomass to Spatial Effects in Pinus kesiya var. langbianensis Forests by Stand Origin and Tree Size"

_forests, doi:10.3390/f15020349_

Round 1
Reviewer 1 Report
Comments and Suggestions for Authors
Spatial regression modelling of biomass equations is a novel idea that fits the scope of the journal. While I think that the introduction needs streamlining before publication, the data, methods and results section are well described and easy to follow. The discussion and conclusions are adequate. The abstract should, however, in my opinion be rewritten. The report delta AIC values are difficult to interpret, and I suggest focusing on delta R² or percent decrease in RMSE. Also, the reported the branch competition thresholds are difficult to understand, without having read the manuscript.
Detailed comments
Line 39: Gradually warm
Fig 2: Distribution of h: nice
Line 178: Please briefly explain the bulk density method
Line 182: What is the average section area method? D²*pi/4*L
Biomass sampling appropriate some of the wording seems akward – language checking
Line 319: N sample mean?
Line 340-342: This sentence is not clear to me
Line 355: RMSE was reduced
Line 399-400: This sentence needs rewording (on the fitted value)
Figure 4,5,6,7: I would use the term “breast height diameter” not “chest height diameter”, since in the figures also the abbreviation “dbh” is used
Line 477-478: This sentence is not clear to me
Line 482-485: Why did you report delta AIC? I think differences in R² and RMSE would be mor informative for the reader; for comparison between models the RMSE difference could be reported in percent.
Line 547: better “growing space”
Line 549-553: How was growing space defined for each organ? How dit the growing spaces differ, how large were they?
Line 575: “by using”
Author Response
Thank you for your careful review of our manuscript. We have answered your questions and revised the manuscript based on your suggestions. Please see the attachment for specific content.

Reviewer 2 Report
Comments and Suggestions for Authors
I think that the paper should be revised before accept. You can find my comments about the paper below:
1-Page 1 line 15. "forest" should be "stand"
2-Page 1 lines 19-20. What the means of these results. What did you use only AIC values for these comparisons in here.
3-Page 1 lines 21-22. These results are expected results.
4-Page 1 lines 30-31. Please change some of the key words. Add new different keywords from title of the paper.
5-Page 2 line 87. Please add "natural and plantation" after forest origin.
6-Page 2 lines 87-98. Plantation and natural stands are including different tree species. I think these situation will effects stand origin and tree size effects
7-Page 4 line 164 Please insert explanation for DBH "diameter breast height"
8-Page 6 Please insert new section for Biomass calculation. This section is a not biomass estimation method.
9-Page 6 lines 204-205 Please give a explanation for selection of a distance-based bandwidth.
10-Page 8 formulae 11-21 How did you decide these base models?
11-Page 8 formulae 11-21 please use same abbreviation for diameter at breast height "D" or "DBH"
12-Page 9 formulae 23 smaller is the better for AIC not close to 0
13-Page 9 formulae 24 "N" should be "n"
14-Page 10 line 335 What did you not use additive models instead of OLS
15-Page 10 Table 1 You used different model forms for same tree component in plantation and natural stands. This situation may be reason of difference results?
16-Table 2 There is no big difference tested models except to OLS
17 Page 17 line 425 please use "tree component" instead of "organ"
Comments on the Quality of English LanguageI think minor editing is required
Author Response

(The authors gave the same response as above.)

Round 2
Reviewer 2 Report
Comments and Suggestions for Authors
I want to thank to the authors for carefully following my suggestions and detailed responses.